# MolTextQA: A Question-Answering Dataset and Benchmark for Evaluating Multimodal Architectures and LLMs on Molecular Structure–Text Understanding

**Siddhartha Laghuvarapu**                                    SL160@ILLINOIS.EDU
*Department of Computer Science*
*University of Illinois at Urbana-Champaign*
*Urbana, IL, USA*

**Namkyeong Lee**                                    NAMKYEONG96@KAIST.AC.KR
*Department of Industrial and Systems Engineering*
*KAIST*
*Daejeon, South Korea*

**Chufan Gao**                                    CHUFAN2@ILLINOIS.EDU
*Department of Computer Science*
*University of Illinois at Urbana-Champaign*
*Urbana, IL, USA*

**Jimeng Sun**                                    JIMENG@ILLINOIS.EDU
*Department of Computer Science*
*Carle Illinois College of Medicine*
*University of Illinois at Urbana-Champaign*
*Urbana, IL, USA*

**Reviewed on OpenReview:** *https://openreview.net/forum?id=tFfqvKE2J5*

**Editor:** Mykola Pechenizkiy

## Abstract

Recent advancements in AI have greatly improved molecular representation learning for property prediction and molecule design. However, leveraging the vast textual molecular data from databases and literature remains challenging. While recent research has explored Large Language Models (LLMs) and multi-modal architectures to link text with molecular structures, existing datasets lack evaluation specificity and comprehensive benchmarking. To address this, we introduce a dataset of 500,000 question-answer pairs covering 240,000 molecules from PubChem, designed for structure-directed questions and text-based molecule retrieval. Moreover, we benchmark various architectural classes fine-tuned using this dataset, including multi-modal architectures, large language models and large reasoning models uncovering several insights. Among the non-LLM baselines, BioT5 and MoleculeSTM achieved the highest performance on the Molecule QA and Molecule Retrieval tasks, respectively, with accuracies approaching 70%. While traditional LLMs struggled with general molecular understanding, our experiments show that fine-tuning LLMs can significantly improve their performance on molecular tasks. Furthermore, large reasoning models, particularly the GPT-o3 series outperform their non-reasoning counterparts and multi-modal architectures, highlighting the importance of explicit reasoning for effective structure–text learning. We have made both the dataset and the fine-tuned models publicly available.

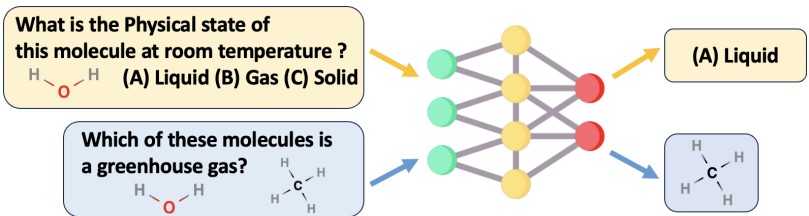

Figure 1: An example of a Question Answering task: In the first question, the objective is to infer certain information from a molecular structure. In the second question, the objective is to retrieve a molecule with properties that satisfy the prompt.

# 1 Introduction

Drug discovery is a costly, labor-intensive process involving extensive validations, laboratory experiments, animal studies, and human trials (Dickson and Gagnon, 2009). A wealth of molecular information exists in textual form within public databases, encompassing drug details (Wishart et al., 2018), toxicity reports (Fonger et al., 2014), extraction methods, physical properties (Kim et al., 2019), patents, chemical reactions (Lowe, 2017), and diverse applications such as materials, fertilizers, and perfumes (Dionisio et al., 2018). While deep learning has signficantly advanced property prediction and computational molecular design Wu et al. (2018), it struggles to extract insights from textual data. Understanding drug side effects, for instance, requires analyzing rich but complex textual sources like FDA reports and clinical trials (U.S. FDA, 2024; CTGov, 2024). Integrating molecular structures with textual information can unlock untapped knowledge, enabling literature-driven discovery of novel materials and drugs with significant scientific and medical potential.

Very recently, there has been a surge in the development of models to decipher the complex relationships between molecular structures and textual descriptions (Liu et al. (2023a,b); Li et al. (2024); Edwards et al. (2022, 2021); Zeng et al. (2022)). Numerous methods have developed multi-modal frameworks, incorporating adaptations of models like CLIP Radford et al. (2021) and BLIP Li et al. (2023), which are specifically designed to learn correlations between visual content and text in the realm of molecular science Liu et al. (2023a,b). Additionally, the emergence of Scientific Large Language Models, such as Galactica Taylor et al. (2022), trained on vast troves of scientific data, represents a significant effort forward in harnessing computational power for molecular understanding and discovery.

Despite advances in model development, challenges in evaluation persist. Existing datasets such as Degtyarenko et al. (2007); Su et al. (2022); Liu et al. (2023a); Fang et al. (2024) often rely on free-form text generation or molecule/text retrieval tasks, using generic prompts like "*Describe the molecule*," which fail to elicit specific molecular properties. Instead, targeted questions, such as "*What is the physical state of the molecule at room temperature?*," would be more effective. Additionally, metrics like BLEU, commonly used in molecule captioning, are inadequate, as vague prompts lead to highly varied responses, making semantic similarity measures unreliable. Further details on these shortcomings are in section 2.2.

In this work, we have developed a comprehensive dataset consisting of over 500,000 question-and-answer (QA) pairs and small molecules represented by SMILES (Simplified Molecular Input

| Dataset | Diverse Questions? | Multiple choice? | Model Benchmark ? | Factual validity ? | Anonymized names ? |
|---|---|---|---|---|---|
| PCDes (Zeng et al., 2022) | ✗ | ✗ | ✗ | ✓ | ✗ |
| CheBI-20 (Edwards et al., 2021) | ✗ | ✗ | ✗ | ✓ | ✓ |
| MoMu (Su et al., 2022) | ✗ | ✗ | ✗ | ✓ | ✗ |
| PubChemSTM (Liu et al., 2023a) | ✗ | ✗ | ✗ | ✓ | ✓ |
| MolInstructions (Fang et al., 2024) | ✗ | ✗ | ✓ | ✓ | ✓ |
| InstructMol (Cao et al., 2023) | ✗ | ✗ | ✓ | ✓ | ✗ |
| 3DMolLM (Li et al., 2024) | ✓ | ✗ | ✗ | ✗ | ✗ |
| MolTextQA (ours) | ✓ | ✓ | ✓ | ✓ | ✓ |

Table 1: **Different molecule captioning or instruction datasets.**

Line Entry System) (Weininger, 1988) sequences. These QAs are crafted from a rich base of textual data sourced from PubChem (Kim et al., 2019), encompassing a wide array of information such as chemical structures, physical properties, applications, and uses of molecules in drugs and biological pathways, as well as manufacturing details. We believe this dataset will significantly enhance the ability to infer information from molecular structures aid in the design of new molecules and improve the capability to evaluate these processes more effectively (See Figure 1). Our main contributions are:

1. We created a comprehensive dataset with about 500,000 QA pairs for 240,000 distinct molecule SMILES sequences across various categories, including multiple-choice answers to improve evaluation precision on specific areas including chemical information, biological information, physical properties, and more.
2. To ensure the reliability of our dataset, we implemented a comprehensive validation process that includes human annotation of a small subset to evaluate data accuracy.
3. We conduct an extensive evaluation of this dataset using state-of-the-art (SoTA) molecule-text multimodal models and recent advancements in large language models. Our analysis provides valuable insights into the advantages and limitations of current models, highlighting their performance in understanding and interpreting molecular data.

To the best of our knowledge, this work represents the first large-scale dataset and benchmarking effort dedicated to diverse directed question-answering methodologies for small molecules.

## 2 Related Work

This section overviews the key related works in molecule-text learning with a single model and separate encoders, language models and various benchmarking datasets. We first describe various models in 2.1, followed by datasets in 2.2.

### 2.1 Models Architectures in Molecule-Text learning

**Molecule-Text learning with a Single Model:** Initial models in molecule-text modeling combined molecules and texts using a unified encoder-only framework. Key works include KV-PLM (Zeng et al., 2022) and GPT-MolBERTa (Balaji et al., 2023), built on the foundations of SciBERT (Beltagy et al., 2019) and RoBERTa (Liu et al., 2019), respectively. KV-PLM fine-tunes SciBERT with SMILES sequences from 15,000 PubChem (Kim et al., 2019) descriptions, employing masked language modeling for representation learning. This model acts as a dual encoder for text and

molecule representations. GPT-MolBERTa, using RoBERTa as its base, integrates descriptions from ChatGPT (OpenAI, 2022), which may introduce non-factual information, affecting reliability.

**Molecule-Text with Separate Encoders:** Recent and more effective models used multi-modal strategies by integrating separate encoders for molecules and text. For instance, Text2Mol (Edwards et al., 2021) employs SciBERT, akin to KV-PLM, for text processing, while molecule representation benefits from Mol2Vec (Jaeger et al., 2018) tokens as initial inputs, utilizing contrastive training to synchronize molecule and text embeddings, mirroring CLIP's (Radford et al., 2021) approach. Conversely, MoMu (Su et al., 2022) introduces molecules as graphs, leveraging a Graph Isomorphism Network (GIN) (Xu et al., 2018) for molecule encoding and uses SciBERT for text processing. MoleculeSTM (Liu et al., 2023a) enhances its methodology by training with both SMILES strings and molecular graphs, initializing SMILES encoder weights from MegaMolBART (NVIDIA, 2021) and adopting a GIN model for graph representation, with text decoding also relying on SciBERT. All these methodologies use of contrastive learning to align text and molecule representations, gaining advantages from expansive datasets and diverse molecular encoding techniques. Alternatively, MolT5 (Edwards et al., 2022) has fine-tuned a T5 model(Raffel et al., 2020) to train separate SMILES and text encoders and decoders. Recently, MolCA (Liu et al., 2023b) utilize the BLIP model (Li et al., 2023), employing a GINE model (Brossard et al., 2020) for graph encoding and SciBERT for text, and text generation fine-tuned by LoRA optimization of the Galactica model (Taylor et al., 2022). 3DMolLM(Li et al., 2024) extended this framework to include question prompts for more directed QA.

**Decoder-based Scientific Large Language Models and Large Reasoning models:** The Galactica model (Taylor et al., 2022), trained explicitly on scientific data, supports a variety of scientific tasks such as generating molecular and protein sequences, question answering, code generation, and mathematical problem-solving. We also evaluate popular large language models (LLMs) like GPT-3.5 (OpenAI, 2022) and LLaMA (Touvron et al., 2023) in our work, noting their strong performance across many tasks, including those in the scientific domain. Moreover, we benchmark recent large reasoning models that demonstrate strong capabilities in multi-step reasoning via chain-of-thought prompting. We expect these models to be particularly effective in capturing the nuanced relationships between molecular structures and their textual descriptions. In particular, we evaluate several models from the GPT family (Achiam et al., 2023) including GPT-4.1, GPT-4o-mini, GPT-o3, and GPT-o4-mini, as well as fine tuned experiments of the open-source Qwen (Qwen et al., 2025) model.

## 2.2 Corresponding Benchmark Datasets for Molecule-Text learning

All the methods discussed so far draw upon datasets sampled from various public sources. For example, KV-PLM introduced PCdes, utilizing PubChem to compile a dataset of 15,000 samples. Additionally, KV-PLM incorporated a set of multiple-choice questions (MCQs) akin to those in this work, albeit with a smaller scope of approximately 1,500 entries, which, in context, was considered a significantly large dataset. Text2Mol developed the CheBI-20 dataset by sampling from the Chemical Entities of Biological Interest (ChEBI) (Degtyarenko et al., 2007) database, resulting in a collection of approximately 20,000 molecular descriptions. MoMu extracted 50,000 captions from PubChem, while MoleculeSTM significantly expanded this approach by sampling over 300,000 captions from PubChem to create the PubChemSTM dataset. 3DMolLM (Li et al., 2024) has curated a set of question-answering pairs from these captions using ChatGPT.

| Splits | Molecules | Total QAs | Physical Properties | Chemical Information | Biological Information | Source | Application |
|---|---|---|---|---|---|---|---|
| **Pretrain** | 213336 | 421227 | 39512 | 183936 | 38757 | 145590 | 13415 |
| **Train** | 20000 | 54842 | 3569 | 28115 | 10448 | 11862 | 848 |
| **Valid** | 2500 | 5754 | 331 | 2809 | 908 | 1620 | 86 |
| **Test** | 5000 | 11922 | 691 | 5941 | 1990 | 3140 | 160 |
| **Total** | 240836 | 493,742 | 45091 | 224468 | 52914 | 165856 | 14670 |

Table 2: **Dataset Statistics:** Distribution of questions across different splits and categories

**Limitations in existing datasets:** We identify several limitations in previous dataset curation methodologies that justify the development of our current dataset and benchmarking efforts. Detailed discussions of these issues, with specific examples, are presented in Appendix.

1. **Lack of Specificity in Prompts and Question Diversity:** Datasets like PubChemSTM(Liu et al., 2023a), MoMu(Su et al., 2022), CheBI-20(Edwards et al., 2021), and PCDes(Zeng et al., 2022) rely on free-form captions generation from PubChem(Kim et al., 2019), which lack task categorization and directed question prompting. Others, like InstructMol(Cao et al., 2023) and MolInstructions(Fang et al., 2024), use generic prompts (e.g., *"Describe the molecule"*) that fail to extract specific molecular information, necessitating more targeted questions.

2. **Evaluation of Text Generation:** Models such as MolT5 (Edwards et al., 2022), MoMu (Su et al., 2022), MolCA (Liu et al., 2023b), and 3DMolLM (Li et al., 2024) generate free-form text from SMILES inputs. When questions lack specificity, evaluation metrics like BLEU and ROUGE struggle to differentiate between diverse responses, such as physical properties versus industrial applications. Moreover, answers like "*The state of the molecule is water*" and "*The stage of the molecule is gas*" may score similarly despite their clear differences. Employing multiple-choice QA formats can mitigate these issues by restricting the output space and enabling more accurate evaluation.

3. **Factual Correctness, Information Leakage, and Dataset Scope:** Some datasets (e.g., 3DMolLM(Li et al., 2024)) use LLM-generated data, raising reliability concerns. Additionally, they fail to anonymize molecule names (e.g., *"What are the physical properties of Aspirin?"*), providing unintended hints. Test sets are biased and are composed of molecules with well-documented captions, overlooking underexplored ones.

4. **Benchmarking Across Model Classes:** Architectural differences among models like MolT5, MoleculeSTM, 3DMolLM, Galactica, and Llama complicate direct comparisons due to varying input formats and training paradigms. We conduct benchmarking across architectural classes in this work.

5. **Benchmarking Large Reasoning Models:** Recent large reasoning models have demonstrated strong performance on complex tasks through chain-of-thought prompting, making them particularly promising for scientific applications. However, existing benchmarks do not evaluate these models in the context of structure–text understanding. The multiple-choice QA format of our benchmark further enables the use of post-training techniques such as Generalized Relative Preference Optimization (GRPO) (Shao et al., 2024), which are well-suited for aligning models with task-specific preferences.

We summarize the differences in these datasets in Table 1. In this work, we introduce MolTextQA, a comprehensive dataset designed to benchmark molecule-text relationship learning. MolTextQA

| Attribute | Data |
|---|---|
| SMILES sequence | CC(=O)C |
| Question | What is the physical state of the molecule at room temperature? |
| Options | (a) Liquid (b) Solid (c) Gas |
| Correct Option | (a) Liquid |
| Sentence | The physical state of the molecule is liquid. |
| SMILES options | (a) CH4 (b) CC(=O)C (c) C(=O)([O-])[O-].[Ca+2] |
| Correct SMILE | (b) CC(=O)C |
| PubChem ID | 180 |
| Category | Physical Properties |

Table 3: **A sample datapoint**

features directed question-answering on small molecules and incorporates multiple-choice questions to enhance accuracy. Additionally, we benchmark molecule-text models across architectural classes to facilitate direct comparisons. It is important to note that this dataset is not intended to replace existing resources and can be used with other datasets to supplement training. The dataset seeks to enhance model evaluation and enable direct comparisons between different methodologies.

## 3 Building the MolTextQA Dataset

### 3.1 Data Source

Our work primarily utilizes the PubChem library (Kim et al., 2019), a resource overseen by the National Center for Biotechnology Information (NCBI) for cheminformatics and drug discovery research. Compiling information from over 750 sources, this dataset supports diverse drug discovery tasks like property prediction and repurposing. Adopting Liu et al.'s approach, we crawl PubChem to extract descriptions and SMILES strings for small molecules, covering aspects from chemical properties to biological effects. This enabled an in-depth analysis of molecule characteristics, enriching our dataset with diverse questions and answers. PubChem is freely available and licensed for non-commercial purposes. Licensing terms are elaborated in the Appendix A.

### 3.2 Dataset overview

The dataset comprises approximately 500,000 questions related to more than 240,000 molecules, categorized into five distinct areas: *Chemical Information*, *Physical Properties*, *Biological Information*, *Source*, and *Application*. Specifics of each category including examples are outlined in the Appendix E. We provide detailed dataset statistics across these QA categories in Table 2. Further, in Table 3, we depict a sample data point indicating different attributes. Each data point includes a question, options, a SMILES sequence, a set of candidate answers, a question category, the correct choice, and options in sentence form.

Each datapoint also includes a set of candidate SMILES randomly sampled from the data and useful for the Molecule Retrieval task. A Tanimoto similarity threshold of 0.2 was applied during sampling to ensure that the selected molecules are sufficiently distinct from one another. The dataset is divided into test, valid, train and pretrain sets for different stages of model development and evaluation.

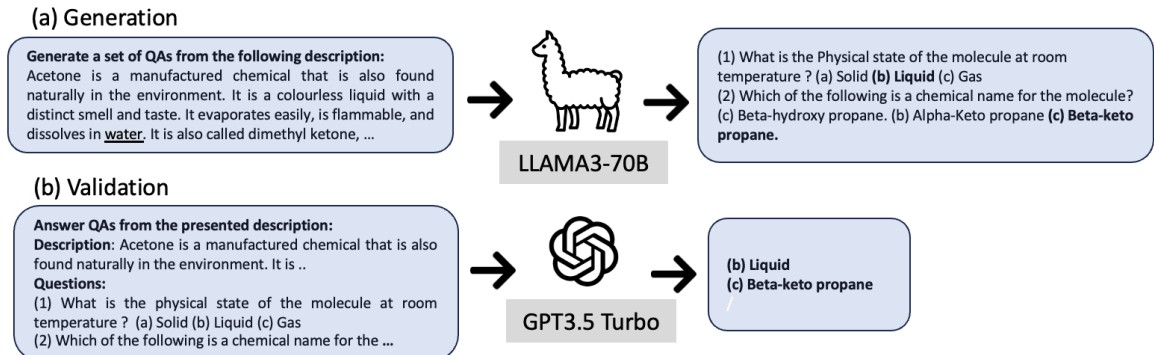

Figure 2: **Procedure for data generation:** (a) molecule captions are used to generate a set of QAs using Llama3-70B. (b) The generated QAs are validated for correctness using GPT 3.5.

## 3.3 Data construction

In this work, we employed different LLMs, specifically Llama3-70B, Llama3-8B, and ChatGPT3.5, for constructing a question-and-answer dataset. The methodology involves passing molecular descriptions to the LLMs and prompting them to generate a set of questions with multiple-choice options that are semantically related to ensure challenging and informative questions. To prevent the possibility of inferring information solely from the molecule's common name, we specifically prompt the LLM to anonymize the common name of the molecule in the resulting QAs. This ensures that downstream models must rely on molecule structure alone for inference. Due to the expensive nature of the Llama3-70B API, the pretrain split was generated with Llama3-7B, while the other splits are generated using the Llama3-70B.

Additionally, we instructed the LLMs to produce a short-sentence answer for each question, to facilitate text-generation tasks. This process is depicted in Figure 2(a). Each generated question was then categorized into one of five distinct categories to streamline the evaluation process. The specifics of the prompts utilized in this process and LLM versions are documented in the Appendix D. It is important to note that LLMs are not utilized as sources of factual information; rather, only to transform data into a different format.

## 3.4 Dataset validation

To enhance the reliability of the generated question-and-answer content and mitigate the risk of fabricated information ("hallucinations") or uninformative, nonsensical questions, we implemented a two-pass validation process. First, following a methodology similar to that outlined by Es et al., we employed an alternative language model (GPT-3.5) to verify the accuracy of the QA pairs. In this step, the caption, question, and multiple-choice options were provided to the model, which was tasked with selecting the correct answer based on the given context. QA pairs where the model either failed to choose the correct answer or could not deduce it from the provided information were excluded from our dataset, ensuring that the content is grounded in the factual information from the captions. In the second stage, the remaining questions were passed to the LLM for an additional verification step. This phase focused on filtering out questions that were unanswerable, uninformative, or not

useful for evaluating molecular characteristics and applications. Detailed descriptions of the prompts used in both stages can be found in the Appendix D. Figure 2 depicts the data generation process.

### 3.4.1 MAJOR FAILURE MODES

The following are the most commonly observed failure modes -

**Hallucinated facts:** The model fabricates a property not present in the PubChem caption (e.g., assigning an incorrect boiling point).

**Irrelevant questions:** The question queries information not inferable from the caption or molecular structure (e.g., FDA approval status).

**Ambiguous options:** Multiple answer choices could reasonably be correct (e.g., "pungent" vs. "unpleasant" odour).

**Name leakage:** The question or an answer option contains the molecule's common name.

### 3.5 Benchmarking Dataset efficacy

To assess the accuracy of language models in generating question-and-answer pairs, we manually reviewed a random sample of 400 QA pairs from the test set. Our evaluation focused on three key criteria: whether the question could be logically derived from the provided caption, the unambiguous correctness of the answer, and the relevance of the question—specifically, ensuring it avoids uninformative queries and contributes to meaningful chemical or biological insights based on the structure. We have found that 391 of the 400 samples satisfy this criteria. This corresponds to a greater than 96.13 percent accuracy on the entire dataset with a p-value of $<0.05$ under a hypergeometric test, indicating statistically significant performance. We elaborate the specifics of the hypothesis test and human verification in Appendix F. We also include these samples with the supplementary material.

## 4 Benchmarking on MolTextQA

This section evaluates the MolTextQA dataset with various models, as detailed in subsection 4.1, which includes both molecule-text multi-modal and scientific language models. For model specifics and training details, see Appendix H. The tasks and objectives for the benchmark are outlined in section 4.2. We then assess model performance in a zero-shot setting (4.3) and 4.4 covers model fine-tuning and their performance evaluation.

### 4.1 Benchmarked Models in Experiments

We benchmark the dataset on the following models

**Single Encoder Architectures: SciBERT** (Beltagy et al., 2019) is a BERT-based model fine-tuned on scientific papers. **KV-PLM** (Zeng et al., 2022) extends SciBERT by pre-training on molecule-text pairs from PubChem, incorporating SMILES fine-tuning with max hinge loss for improved retrieval.

**Separate Encoder Architectures: MoleculeSTM** (Liu et al., 2023a) employs a dual-encoder combining SciBERT (Beltagy et al., 2019) and a Graph Isomorphism Network (GIN) (Xu et al., 2018) to encode text and chemical structures, using PubChem data and InfoNCE loss for molecule-text alignment. **MoMu** (Su et al., 2022) is a similar model that uses 3D structure information.

**Large Language Models (Decoder-Only): Llama** (Touvron et al., 2023) os an autoregressive transformer models fine-tuned for instruction adherence, demonstrating versatility across tasks like

QA and code generation. We experiment with Llama2-7B, Llama2-70B, Llama3-8B, and Llama3-70B. **GPT-3.5 Turbo** (OpenAI, 2022), from the GPT series, is optimized for human alignment and excels across diverse tasks. **Galactica** (Taylor et al., 2022), an autoregressive decoder-only model trained on scientific content, is effective for specialized biomedical datasets; we evaluate its 125M, 1.3B, and 6.7B versions.

**Reasoning Models: GPT-4.1**, **GPT-4o-mini**, **o3**, and **o4-mini** are recent decoder-only models from the GPT series (Achiam et al., 2023) that demonstrate strong reasoning capabilities via chain-of-thought prompting. These models are optimized for multi-step inference and perform well on structured scientific tasks. We also evaluate open-source **Qwen-2.5** (Qwen et al., 2025) models (0.6B, 4B, 8B), which are instruction-tuned language models designed for general-purpose use. We additionally fine-tune Qwen-0.6B and Qwen-4B using GRPO to improve alignment with the multiple-choice QA format in our benchmark.

**Encoder-Decoder models: MolT5** (Edwards et al., 2022) is a T5 model fine-tuned for molecule captioning, utilizing an encoder-decoder architecture for SMILES and text. **BioT5** (Pei et al., 2023) is a T5 model trained on a larger dataset encompassing small molecules and proteins for enhanced multi-modal performance. **BioT5 Plus** (Pei et al., 2024) further updates BioT5 with additional data sources and multi-task instruction tuning.

## 4.2 Tasks and Objectives

1. **Molecule QA task**: Tests a model's ability to identify molecular properties from a given SMILES string or molecular graph. Since encoder-only models cannot be directly prompted to generate text, we append the question and each answer choice to create a complete sentence, then perform semantic similarity matching to identify the correct option to a SMILES input. Decoder-only models are directly prompted with the question and SMILES input.

2. **Molecule Retrieval task:** Requires models to identify the correct SMILES string corresponding to a textual molecule description. Decoder-only LLMs are prompted to select the appropriate SMILES from candidate options, while encoder models retrieve matching structures based on semantic similarity of the embeddings.

Performance is measured by Accuracy:

$$\text{Accuracy} = \frac{\text{Number of Correct Predictions}}{\text{Total Number of Predictions}}$$

## 4.3 Results on zero-shot inference

The results for the Molecule QA and Molecule Retrieval tasks under zero-shot conditions are outlined in Table 4. These evaluations involved models that were not trained on the question-answering dataset introduced in this study. However, models such as MoleculeSTM, MoMu, and KV-PLM, which were trained using a dataset similar to the one presented here, which may partly explain their effectiveness. To avoid data leakage, MoleculeSTM and MoMu were retrained, explicitly excluding test samples from their training sets. For the LLMs, the public checkpoints were prompted with QA tasks; details of these prompts are provided in the Appendix D. Results for the Galactica, MolT5, and BioT5 models are omitted in zero-shot scenarios as they are not aligned for question-answering tasks.

| Model | Entire dataset | Physical Prop. | Chemical Info | Biological Info | Sources | Uses |
|---|---|---|---|---|---|---|
| **Molecule QA** | | | | | | |
| SciBERT | 21.3 ± 0.7 | 20.2 ± 0.7 | 21.5 ± 0.7 | 19.0 ± 0.7 | 22.4 ± 0.8 | 22.5 ± 0.8 |
| KV-PLM | 29.9 ± 0.8 | 27.9 ± 0.8 | 32.0 ± 0.8 | 26.6 ± 0.8 | 28.5 ± 0.8 | 26.7 ± 0.8 |
| MoleculeSTM | 44.7 ± 0.9 | 30.2 ± 0.8 | 48.0 ± 0.9 | 30.9 ± 0.9 | 51.3 ± 0.9 | 31.0 ± 0.9 |
| MoMu | 44.9 ± 0.9 | 28.8 ± 0.8 | 49.6 ± 0.9 | 30.1 ± 0.8 | 50.3 ± 0.9 | 27.8 ± 0.8 |
| GPT-3.5 | 40.3 ± 0.9 | 44.7 ± 0.9 | 38.5 ± 0.9 | 43.3 ± 0.9 | 40.2 ± 0.9 | 47.6 ± 0.9 |
| LLaMA-3 8B | 16.2 ± 0.6 | 20.0 ± 0.7 | 16.1 ± 0.7 | 17.3 ± 0.7 | 14.1 ± 0.6 | 27.3 ± 0.7 |
| LLaMA-3 70B | 58.2 ± 0.9 | 63.3 ± 0.9 | 57.1 ± 0.9 | 56.9 ± 0.9 | 59.6 ± 0.9 | 66.3 ± 0.9 |
| LLaMA-2 7B | 24.6 ± 0.8 | 22.7 ± 0.8 | 25.3 ± 0.8 | 24.9 ± 0.8 | 23.5 ± 0.8 | 24.6 ± 0.8 |
| LLaMA-2 70B | 28.8 ± 0.8 | 30.6 ± 0.8 | 27.1 ± 0.8 | 36.2 ± 0.9 | 26.4 ± 0.8 | 37.4 ± 0.9 |
| Random | 20.7 ± 0.7 | 20.6 ± 0.7 | 22.8 ± 0.7 | 21.1 ± 0.7 | 20.2 ± 0.7 | 19.5 ± 0.7 |
| **Molecule Retrieval** | | | | | | |
| SciBERT | 21.2 ± 0.7 | 21.4 ± 0.7 | 21.9 ± 0.7 | 21.1 ± 0.7 | 19.8 ± 0.7 | 23.0 ± 0.7 |
| KV-PLM | 48.5 ± 0.9 | 47.8 ± 0.9 | 60.0 ± 0.9 | 43.1 ± 0.8 | 30.5 ± 0.8 | 54.0 ± 0.9 |
| MoleculeSTM | 67.4 ± 0.9 | 49.1 ± 0.8 | 77.0 ± 0.9 | 53.0 ± 0.9 | 63.6 ± 0.9 | 54.6 ± 0.9 |
| MoMu | 66.0 ± 0.8 | 45.8 ± 0.8 | 76.5 ± 0.9 | 51.1 ± 0.9 | 61.5 ± 0.9 | 50.3 ± 0.8 |
| GPT-3.5 | 38.3 ± 0.9 | 39.9 ± 0.9 | 47.0 ± 0.9 | 31.1 ± 0.9 | 26.4 ± 0.8 | 36.9 ± 0.9 |
| LLaMA-3 8B | 21.0 ± 0.7 | 22.2 ± 0.7 | 21.9 ± 0.8 | 20.1 ± 0.7 | 19.7 ± 0.7 | 16.0 ± 0.7 |
| LLaMA-3 70B | 52.7 ± 0.9 | 41.3 ± 0.9 | 70.3 ± 0.9 | 38.1 ± 0.8 | 32.6 ± 0.8 | 39.6 ± 0.9 |
| LLaMA-2 7B | 18.6 ± 0.7 | 19.4 ± 0.7 | 19.2 ± 0.7 | 17.6 ± 0.7 | 18.0 ± 0.7 | 16.0 ± 0.7 |
| LLaMA-2 70B | 20.4 ± 0.7 | 16.6 ± 0.7 | 21.9 ± 0.7 | 18.2 ± 0.7 | 19.9 ± 0.7 | 15.5 ± 0.7 |
| Random | 20.3 ± 0.7 | 20.4 ± 0.7 | 21.1 ± 0.7 | 19.8 ± 0.7 | 20.6 ± 0.7 | 19.2 ± 0.7 |

Table 4: **Zero-Shot Setting Accuracy**. Molecule QA requires choosing the correct answer from a SMILES + question. Molecule Retrieval selects the correct SMILES from a property description. Confidence intervls are bootstrapped at 95%

In the Molecule QA task, which requires answering questions based on SMILES inputs, both decoder-only LLMs and multimodal architectures exhibited similar performance. Here, Llama3-70B emerged as the top performer. MoleculeSTM, MoMu, and GPT3.5 showed comparable results. In contrast, smaller models such as Llama-7b and SciBERT approximated random guessing performance, highlighting their limited applicability. Notably, LLMs performed better in predicting physical properties and uses (e.g., appearance, odor), whereas multimodal architectures excelled in processing chemical information. This suggests that LLMs are adept at handling diverse data types, while multimodal systems effectively leverage structural data inherent in molecular representations useful for inferring chemical information.

For the Molecule Retrieval task, multimodal architectures, particularly MoleculeSTM, significantly outperformed LLMs, accurately retrieving SMILES strings in over 66% of instances. The

marginal superiority of MoleculeSTM over MoMu suggests that 3D pretraining initialization may enhance its retrieval capabilities. Multimodal architectures consistently demonstrated higher accuracy in identifying chemical properties compared to other types similar to the MoleculeQA task. Among LLMs, only Llama3-70B achieved notable performance, with an accuracy exceeding 50%.

## 4.4 Results on Finetuned models

In this section, we discuss the outcome of finetuning our models in Table 5 using the training subset in the proposed dataset. We finetuned the Llama3-8B, Llama2-7B, Galactica and MolT5 models, alongside the top multimodal architectures, MoMu and MoleculeSTM, by fine-tuning them with this subset. For MolT5, we fine-tuned both the pretrained checkpoint and the checkpoints specifically for SMILES generation or caption generation. We did not fine-tune larger LLMs due to the high costs associated with training. We faced challenges in finetuning the architecture proposed in 3DMolLM (Li et al., 2024), which are discussed in the Appendix H.

Upon fine-tuning, all LLMs have exhibited reasonable performance in the molecule QA task. Most notably, the Llama3-8B model showed a 45% improvement in accuracy, surpassing the best performing zero-shot Llama3-70B model from the previous section. The BioT5 model emerged as the best performer overall with a 75% accuracy, suggesting that they can be effectively fine-tuned for question answering tasks. However, this also highlights room for improvement, which future works should focus on. Additionally, the size of the LLMs appears to be advantageous, indicating that scaling could further enhance performance. The results overall indicate that LLMs can be effective in infering molecular properties. On the other hand, multimodal architectures also demonstrated a considereable improvement across categories, increasing performance by 20%. However, the MolT5 models, while outperforming random benchmarks, lagged behind other models. The architecture of MolT5 is similar to BioT5; however, the notable performance gap between them highlights the advantages of BioT5's multi-modal training approach.

For the Molecule Retrieval tasks, all LLMs performed close to randomly, indicating that this architecture might not be well-suited for molecule generation tasks, though it can be useful for inferring properties. In contrast, multimodal architectures consistently outperformed LLMs. Notably, while these architectures showed superior performance in chemical properties in zero-shot settings, fine-tuning enabled them to excel across categories. Despite significant gains in tasks like Physical properties (by over 10%), this has also led to a slight decline in performance for Chemical properties and consequently, overall. This performance dip suggests that embeddings may not fully capture diverse types of information, leading to trade-offs. We also observe a similar trend between BioT5 and BioT5-plus, where BioT5-plus demonstrates improved performance in the "Sources" and "Uses" categories but shows a decline in accuracy for "Physical" and "Chemical Properties." This highlights the need for improved modeling strategies to achieve robust performance across all tasks.

It is important to note that these fine-tuning efforts utilized only a small portion of the full dataset, raising questions about potential outcomes if the entire dataset were employed. Given the success of LLMs in Molecule QA tasks and multimodal architectures in Molecule Retrieval tasks, we speculate that an architecture fine-tuned on QA tasks, which integrates elements from both types of architectures and employs strategies to capture diverse sorts of information, could excel in both scenarios. Exploring this possibility will be a focus of our future research.

| Model | Entire dataset | Physical Properties | Chemical Info | Biological Info | Sources | Uses |
|---|---|---|---|---|---|---|
| **Molecule QA** | | | | | | |
| MoleculeSTM | 65.1 ± 0.9 | 68.6 ± 0.9 | 61.9 ± 0.9 | 65.4 ± 0.9 | 69.9 ± 0.9 | 71.1 ± 0.9 |
| MoMu | 65.1 ± 0.8 | 70.8 ± 0.8 | 60.7 ± 0.8 | 66.7 ± 0.8 | 70.6 ± 0.8 | 71.7 ± 0.8 |
| LLaMA-3 8B | 60.4 ± 0.9 | 64.3 ± 0.9 | 58.7 ± 0.9 | 64.1 ± 0.9 | 60.7 ± 0.9 | 55.1 ± 0.9 |
| LLaMA-2 7B | 41.8 ± 0.9 | 42.1 ± 0.9 | 44.0 ± 0.9 | 41.6 ± 0.9 | 38.2 ± 0.9 | 37.4 ± 0.9 |
| Galactica-125M | 44.0 ± 0.9 | 43.4 ± 0.9 | 43.1 ± 0.9 | 43.6 ± 0.9 | 46.3 ± 0.9 | 37.4 ± 0.9 |
| Galactica-1.3B | 61.0 ± 0.8 | 62.6 ± 0.8 | 58.6 ± 0.8 | 62.4 ± 0.8 | 64.8 ± 0.8 | 48.7 ± 0.8 |
| Galactica-6.7B | 69.0 ± 0.8 | 70.4 ± 0.8 | 65.7 ± 0.8 | 73.0 ± 0.8 | 72.5 ± 0.8 | 65.2 ± 0.8 |
| MolT5-large | 34.2 ± 0.8 | 30.6 ± 0.8 | 34.3 ± 0.8 | 38.3 ± 0.8 | 32.6 ± 0.8 | 26.7 ± 0.8 |
| MolT5-large-s2c | 34.7 ± 0.9 | 47.3 ± 0.9 | 31.2 ± 0.9 | 37.9 ± 0.9 | 36.5 ± 0.9 | 31.6 ± 0.9 |
| BioT5 | 75.1 ± 0.9 | 81.0 ± 0.9 | 73.7 ± 0.9 | 73.8 ± 0.9 | 77.5 ± 0.9 | 68.5 ± 0.9 |
| BioT5-plus | 71.2 ± 0.9 | 69.4 ± 0.9 | 68.1 ± 0.9 | 72.6 ± 0.9 | 78.4 ± 0.9 | 72.7 ± 0.9 |
| Random | 20.7 ± 0.7 | 20.6 ± 0.7 | 22.8 ± 0.7 | 21.1 ± 0.7 | 20.2 ± 0.7 | 19.5 ± 0.7 |
| **Molecule Retrieval** | | | | | | |
| MoleculeSTM | 65.3 ± 0.9 | 60.0 ± 0.9 | 72.4 ± 0.9 | 54.6 ± 0.9 | 60.2 ± 0.9 | 62.0 ± 0.9 |
| MoMu | 63.6 ± 0.9 | 56.3 ± 0.9 | 70.5 ± 0.9 | 53.3 ± 0.9 | 59.2 ± 0.9 | 57.8 ± 0.9 |
| LLaMA-3 8B | 20.6 ± 0.7 | 19.8 ± 0.7 | 20.7 ± 0.7 | 20.1 ± 0.7 | 20.9 ± 0.7 | 22.5 ± 0.7 |
| LLaMA-2 7B | 20.6 ± 0.7 | 20.4 ± 0.7 | 20.0 ± 0.7 | 20.8 ± 0.7 | 21.5 ± 0.7 | 21.4 ± 0.7 |
| Galactica-125M | 21.6 ± 0.7 | 28.0 ± 0.7 | 21.6 ± 0.7 | 20.9 ± 0.7 | 20.1 ± 0.7 | 31.0 ± 0.7 |
| Galactica-1.3B | 22.2 ± 0.7 | 29.6 ± 0.7 | 22.4 ± 0.7 | 21.7 ± 0.7 | 19.7 ± 0.7 | 31.0 ± 0.7 |
| Galactica-6.7B | 22.3 ± 0.7 | 30.4 ± 0.7 | 22.6 ± 0.7 | 22.2 ± 0.7 | 19.3 ± 0.7 | 33.2 ± 0.7 |
| MolT5-large | 23.5 ± 0.8 | 39.8 ± 0.8 | 23.9 ± 0.8 | 23.4 ± 0.8 | 18.2 ± 0.8 | 41.2 ± 0.8 |
| MolT5-large-c2s | 23.0 ± 0.8 | 32.3 ± 0.8 | 23.9 ± 0.8 | 21.3 ± 0.8 | 19.9 ± 0.8 | 30.0 ± 0.8 |
| BioT5 | 23.3 ± 0.8 | 37.6 ± 0.8 | 21.3 ± 0.8 | 21.1 ± 0.8 | 17.3 ± 0.8 | 33.2 ± 0.8 |
| BioT5-plus | 22.3 ± 0.8 | 31.2 ± 0.8 | 21.3 ± 0.8 | 19.5 ± 0.8 | 17.8 ± 0.8 | 27.7 ± 0.8 |
| Random | 20.3 ± 0.7 | 20.4 ± 0.7 | 21.1 ± 0.7 | 19.8 ± 0.7 | 20.6 ± 0.7 | 19.2 ± 0.7 |

Table 5: **Finetuning performance:** Accuracy of different models in the finetuning setting for both Molecule QA and Molecule Retrieval tasks, reported with bootstrapped 95% confidence intervals.

## 4.5 Results on Large Reasoning Models

In this section, we benchmark recent Large Reasoning models that demonstrate advanced reasoning capabilities via chain-of-thought prompting. These models have shown strong performance across domains involving multi-step reasoning, and we hypothesize that such capabilities can be beneficial in understanding the nuanced relationships between molecular structure and text. The multiple-choice QA format of our benchmark also enables training these models using reinforcement learning strategies such as GRPO, which we explore in this work.

| Model | Entire dataset | Physical Properties | Chemical Info | Biological Info | Sources | Uses |
|---|---|---|---|---|---|---|
| **Molecule QA** | | | | | | |
| GPT-4-1 | 48.4 ± 0.9 | 46.8 ± 0.9 | 39.2 ± 0.9 | 37.5 ± 0.9 | 38.2 ± 0.9 | 34.5 ± 0.9 |
| GPT-4o-mini | 47.8 ± 0.9 | 53.5 ± 0.9 | 51.8 ± 0.9 | 37.9 ± 0.9 | 46.3 ± 0.9 | 30.5 ± 0.9 |
| o3 | 60.3 ± 0.9 | 61.4 ± 0.9 | 73.6 ± 0.9 | 53.3 ± 0.9 | 44.4 ± 0.9 | 42.9 ± 0.9 |
| o4-mini | 57.9 ± 0.9 | 51.8 ± 0.9 | 70.2 ± 0.9 | 49.5 ± 0.9 | 42.3 ± 0.9 | 48.2 ± 0.9 |
| Qwen-0.6B (base) | 31.8 ± 0.8 | 39.5 ± 0.8 | 30.6 ± 0.8 | 34.3 ± 0.8 | 30.9 ± 0.8 | 30.8 ± 0.8 |
| Qwen-4B (base) | 45.2 ± 0.9 | 57.8 ± 0.9 | 45.5 ± 0.9 | 50.1 ± 0.9 | 38.5 ± 0.9 | 41.4 ± 0.9 |
| Qwen-8B (base) | 43.0 ± 0.9 | 62.5 ± 0.9 | 46.7 ± 0.9 | 42.7 ± 0.9 | 36.8 ± 0.9 | 42.9 ± 0.9 |
| Qwen-0.6B (GRPO) | 53.8 ± 0.9 | 55.0 ± 0.9 | 50.1 ± 0.9 | 56.3 ± 0.9 | 58.9 ± 0.9 | 56.7 ± 0.9 |
| Qwen-4B (GRPO) | 64.0 ± 0.9 | 64.1 ± 0.9 | 54.1 ± 0.9 | 60.2 ± 0.9 | 46.7 ± 0.9 | 53.4 ± 0.9 |
| **Molecule Retrieval** | | | | | | |
| GPT-4-1 | 62.3 ± 0.9 | 60.6 ± 0.9 | 77.5 ± 0.9 | 60.0 ± 0.9 | 32.6 ± 0.9 | 53.4 ± 0.9 |
| GPT-4o-mini | 45.4 ± 0.9 | 37.4 ± 0.9 | 63.3 ± 0.9 | 32.8 ± 0.9 | 22.6 ± 0.9 | 21.0 ± 0.9 |
| o3 | 70.3 ± 0.9 | 72.3 ± 0.9 | 82.6 ± 0.9 | 64.2 ± 0.9 | 53.5 ± 0.9 | 52.0 ± 0.9 |
| o4-mini | 67.9 ± 0.9 | 60.7 ± 0.9 | 81.2 ± 0.9 | 58.9 ± 0.9 | 51.1 ± 0.9 | 59.1 ± 0.9 |
| Qwen-0.6B (base) | 22.1 ± 0.7 | 20.0 ± 0.7 | 22.8 ± 0.7 | 23.9 ± 0.7 | 19.9 ± 0.7 | 15.8 ± 0.7 |
| Qwen-4B (base) | 49.0 ± 0.9 | 50.5 ± 0.9 | 62.6 ± 0.9 | 40.7 ± 0.9 | 27.4 ± 0.9 | 43.3 ± 0.9 |
| Qwen-8B (base) | 48.0 ± 0.9 | 31.3 ± 0.9 | 67.5 ± 0.9 | 52.6 ± 0.9 | 25.3 ± 0.9 | 14.3 ± 0.9 |
| Qwen-0.6B (GRPO) | 29.5 ± 0.8 | 33.0 ± 0.8 | 33.8 ± 0.8 | 29.7 ± 0.8 | 19.7 ± 0.8 | 34.5 ± 0.8 |
| Qwen-4B (GRPO) | 58.9 ± 0.9 | 60.0 ± 0.9 | 73.0 ± 0.9 | 51.2 ± 0.9 | 36.1 ± 0.9 | 51.4 ± 0.9 |

Table 6: **Reasoning model performance:** Accuracy of reasoning-capable language models in zero-shot and GRPO fine-tuned settings, evaluated on both Molecule QA and Molecule Retrieval tasks. Reported with bootstrapped 95% confidence intervals.

We evaluate four models from the GPT series (GPT-4-1, GPT-4o-mini, o3, and o4-mini) and three checkpoints from the open-source Qwen series (0.6B, 4B, 8B) in a zero-shot setting. Additionally, we fine-tune the Qwen-0.6B and Qwen-4B models using GRPO, assigning a reward of +1 to the correct answer and 0 otherwise. This is, to our knowledge, the first use of online RL for molecule-focused QA at scale. These results are shown in Table 6.

Overall, the o3 model achieved the highest performance in the Molecule QA task, outperforming even some fine-tuned baselines. The GRPO-finetuned Qwen-4B model also performed competitively, indicating that preference-based reinforcement learning can enhance molecular reasoning. For the Molecule Retrieval task, GPT-based reasoning models notably outperformed prior LLMs, with o3 and o4-mini achieving strong performance across all categories. Fine-tuning Qwen-4B results in notable improvements, particularly in the Chemical and Biological Information categories, indicating that reasoning based models can be effectively adapted to molecular domains.

These results highlight the potential of reasoning-oriented LLMs for both QA and retrieval-style tasks, especially when combined with preference optimization strategies. All results were obtained without using the full dataset and only on the training set, and we anticipate further gains with larger-scale training.

## 5 Conclusion

Our work introduces MolTextQA, a novel dataset featuring over 500,000 QA pairs related to small molecule structures, covering a broad spectrum of molecular properties and applications. We have benchmarked MolTextQA against a diverse array of large language models and state-of-the-art multimodal architectures, analyzing their strengths and weaknesses. We also see potential in extending our approach to include other scientific modalities, like proteins, to widen the dataset's applicability. We aspire for MolTextQA to become a foundational resource for the development of more efficient molecule-text foundation models. We anticipate its application in drug discovery, materials science, and other fields.

## 6 Limitations

We acknowledge a few limitations in the dataset creation process. First, the dataset exhibits minor inaccuracies, as reported in Section 3.5, which could potentially complicate model training. Future work will focus on implementing more rigorous validation strategies to enhance data precision. Additionally, the dataset includes a broad spectrum of questions, a small fraction of which may appear straightforward, such as identifying the polarity of a molecule—an aspect that only requires a basic knowledge of chemistry. While this diversity of questions benefits model training, it also suggests we could refine our selection process to ensure each question more effectively serves the dataset's purpose. Furthermore, the dataset covers data across a wide range of general categories. The exploration of data acquisition and fine-tuning of specialized datasets for niche applications is an area that requires further investigation. Finally, because the dataset is sourced from PubChem, it may inherit biases favoring well-studied or extensively documented molecules. Addressing such representational biases remains an important direction for future research.

## 7 Broader Impact Statement

Our work contributes to the advancement of molecule-text relationship learning by introducing MolTextQA, a structured question-answering dataset designed to improve the precision of molecular inference. The dataset addresses gaps in existing resources by providing multiple-choice and textual answers based on molecular inputs, facilitating better benchmarking and evaluation of AI-driven cheminformatics models.

### 7.1 Scientific Contributions

MolTextQA supports applications in drug discovery, retrosynthesis, and materials science by providing a benchmark for assessing molecular reasoning capabilities in AI models. The structured QA format enables more precise evaluation compared to existing datasets, facilitating improvements in molecular representation learning. We also conduct a thorough evaluation of models on MolTextQA

to assess their ability to generalize across diverse properties. This evaluation highlights the strengths and limitations of existing approaches, guiding future research toward more effective models.

## 7.2 Ethical Considerations

All data, code, and models used in this research are sourced from publicly available domains that permit free distribution, ensuring compliance with ethical standards of transparency and accessibility. Details regarding the licensing of the dataset source (PubChem) are provided in Appendix A. Given the long-standing traditions within the biomedical and cheminformatics communities, our work adheres to established norms, and we do not anticipate any ethical concerns specific to this study. However, we acknowledge the potential for dual-use applications, including the accelerated design of harmful or environmentally hazardous compounds. We emphasize the importance of institutional oversight and audit mechanisms when deploying such models. To promote transparency and reproducibility, we have made the dataset publicly available, as described in Appendix A.

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

## Appendix A. Dataset Description

The MolTextQA dataset is publicly available and hosted by Huggingface, ensuring permanent availability due to its DOI identifier. The dataset can be accessed via the following links:

- **Dataset Link**: `https://huggingface.co/datasets/sl160/MolTextQA`

- **DOI**: 10.57967/hf/2443

- **Croissant Metadata URL**: `https://huggingface.co/api/datasets/sl160/MolTextQA/croissant`

The dataset is organized in a CSV file format, which is a standard and widely used format in machine learning applications. Instructions for reading and manipulating the dataset are provided at the dataset link.

### A.1 Dataset Structure

Each data point in the dataset contains the following fields:

- **CID**: The PubChem Identifier of the molecule.

- **QID**: The identifier of the question within a CID.

- **Category**: The category of the data point, following this convention:

  1. Physical properties
  2. Chemical information
  3. Biological uses
  4. Sources
  5. General applications

- **Sentence**: A sentence summary of the question and answer.

- **Question**: The actual question asked.

- **Options**: A set of options for the answer, of which one is correct.

- **Correct_option**: The index (1-based) of the correct option.

- **Retrieval_options**: A set of PubChem IDs used for molecule retrieval from the sentence task.

- **Retrieval_correct**: The correct option in the retrieval task.

For benchmarking and further details regarding the application of this dataset in machine learning tasks, please refer to the project repository. The model weights will be made available upon acceptance and all the results are reproducible. The predictions file and the codes used for generating the results in the benchmark are also available in the repository.

- Dataset and Benchmark Repository: `https://github.com/siddharthal/MolTextQA/`

## A.2 Intended Uses of the Dataset

The dataset is primarily intended to be used for molecule-text relationship learning. The task of molecule-text learning has been gaining increasing attention in recent research. However, the current datasets and developed models do not enable structured inference, and evaluation is not precise. The MolTextQA dataset addresses these challenges by offering a question-answering format with multiple-choice answers. Questions are based on a small molecule input, with answers provided in textual sentence or multiple-choice format. The dataset is intended for applications in fields such as drug discovery, retrosynthesis, and the discovery of materials like fertilizers, pesticides, and perfumes.

## A.3 License

The MolTextQA dataset will be distributed under the Creative Commons Attribution 4.0 International (CC BY 4.0) license, which permits use, distribution, and reproduction in any medium, provided the original work is properly cited.

## A.4 Rights and Responsibilities

The authors bear all responsibility in case of violation of rights associated with the dataset.

## Appendix B. Dataset source

The primary source for building this dataset is PubChem (Kim et al., 2019). PubChem consolidates chemical data from multiple public sources. A comprehensive list of these sources is accessible at `https://pubchem.ncbi.nlm.nih.gov/sources/`. The licensing terms on PubChem are stated as: "Works produced by the U.S. government are not subject to copyright protection in the United States. Any such works found on National Library of Medicine (NLM) Web sites may be freely used or reproduced without permission in the U.S. Please acknowledge NLM as the source of the information by including the phrase "Courtesy of the National Library of Medicine" or "Source: National Library of Medicine." More details on the licensing terms can be found at `https://www.nlm.nih.gov/web_policies.html`.

For efficient data retrieval, the PubChem Power User Gateway offers abstracts of compound records in XML format. This facilitates the extraction and analysis of chemical information by enabling users to search for molecular descriptions and their unique PubChem Compound Identifier (CID). This CID is then used to fetch the Simplified Molecular-Input Line-Entry System (SMILES) representation for each compound listed in PubChem. For utilizing the PubChem Power User Gateway, visit `https://pubchem.ncbi.nlm.nih.gov/docs/power-user-gateway`. This approach for obtaining PubChem data is also followed by (Fang et al., 2024; Cao et al., 2023; Li et al., 2024), all publicly available.

## Appendix C. Limitations of existing datasets

In this section, we discuss some limitations of existing datasets, illustrating each point with concrete examples.

**Lack of Specificity in Prompts and Question Diversity:** Existing datasets such as PubChem-STM (Liu et al., 2023a), MoMu (Su et al., 2022), CheBI-20 (Edwards et al., 2021), and PCDes (Zeng et al., 2022) predominantly consist of captions derived from PubChem (Kim et al., 2019). These captions are free-form and include diverse information across various tasks without specific categorization. For instance, consider the following captions from PubChem:

- Caption A: *This molecule is a natural product found in Carica papaya.*

- Caption B: *It is an N-glycosyl compound, a ribose triphosphate, a pyrimidone, and an aminopyrimidine.*

Datasets such as InstructMol (Cao et al., 2023) and MolInstructions (Fang et al., 2024), which also source their captions from similar databases, pose queries like *'Describe the molecule.'* Given the broad range of information in the captions—from molecule manufacturing, to physical properties, to chemical structures, to drug toxicity, to industrial applications—the queries remain insufficiently structured for detailed inference. In contrast, our proposed dataset includes specific queries such as *Is this molecule denser than water?*, *Does this molecule contain a mannose ring?*, or *Is this molecule an antibiotic or an analgesic?*

This issue also affects retrieval models such as MoleculeSTM or MoMu, where the challenge is compounded by the necessity for a single molecule embedding to retrieve both Caption A and Caption B. This task is difficult as these captions semantically reside in distinct spaces.

**Factual Correctness:** Datasets such as 3DMolLM(Li et al., 2024) employ data augmentation from LLMs in their data generation procedure, raising concerns about the overall accuracy of the dataset. Moreover, these datasets generate five questions for each data point, irrespective of whether sufficient information exists. There is no validation process to determine the reliability of the data, leading to the generation of numerous unreliable questions. Consider the following example from PubChem: `https://pubchem.ncbi.nlm.nih.gov/compound/10008613`

The provided caption is:

$(1S, 2S, 4R, 5R, 6R, 9R, 10S, 11R, 12R, 16R, 18S, 21R) - 2, 9, 10, 11 - tetrahydroxy - 4, 6, 12, 17, 17 - pentamethyl - 18 - [(2S, 3R, 4S, 5R) - 3, 4, 5 - trihydroxyoxan - 2 - yl]oxyhexacyclo[11.9.0.0^1, 21.0^4, 12.0^5, 10.0^1 6, 21]docos - 13 - en - 8 - one$ is a natural product found in Actaea yunnanensis and Actaea cimicifuga with data available.

The generated questions in the 3DMolLM dataset are:

- *What is the SMILES code of the molecule?*, which is trivial as SMILES sequence is part of the input.

- *What is the chemical name of this molecule?*,

- *What are some of the functional groups present in this molecule?*,

- *What are the physical properties of this molecule?*, speculating about properties not detailed in the caption.

- *What is the potential biological significance of this molecule?*, hypothesizing about the biological activity without supporting data.

These questions illustrate the challenge of relying on LLM-generated content without appropriate validation, leading to questions that speculate beyond available data.

In contrast, our current work employs a two-stage procedure to validate and filter data generated with LLMs. We also manually evaluate the dataset and estimate its overall accuracy at less than 0.05. Since not much information is available on most molecules, our dataset averages about two questions per molecule, enhancing factual reliability.

**Information Leakage:** The 3DMolLM dataset does not anonymize molecule names, resulting in many questions that include either the common or chemical name of the molecule, thereby providing unintended hints. For example, a question in the dataset - *"What is the main component of Lobaplatin that gives it its anticancer properties?"* offers additional clues that may influence the evaluation process, complicating assessing a model's ability to learn from the molecule sequence or structure alone.

**Evaluation Limited to Samples with Available Data:** Datasets like MoMu, 3DMolLM, and MolCA limit their test sets to molecules accompanied by captions of more than 20 words. This approach inherently biases the evaluation towards well-known and extensively studied molecules. Conversely, the dataset presented in this work, while including these well-documented data points, also augments the test set with molecule captions chosen randomly, not by length. This strategy helps to provide a more balanced and representative evaluation of the model's capabilities across a wider range of molecular data.

**Benchmarking Across Several Model Classes:** Existing datasets often exhibit limitations in their benchmarking scope. For instance, MolInstructions has been evaluated solely using LLMs, while 3DMolLM employs only Llama as an additional baseline. Other models like KV-PLM and MoMu are benchmarked against only a subset of available models. Similarly, MoleculeSTM and MolT5 lack direct comparisons in their evaluations. In contrast, our approach aims to extensively benchmark and compare models across different architectural classes. This broader evaluation is facilitated by the structure of our dataset, which includes questions and multiple-choice options, allowing for a more comprehensive assessment of model performance.

## Appendix D. Prompts for data generation and inference

In this section, we discuss the various prompts for large language models in this work

- In Figure 3, we depict the prompt used to generate QA data (section 3.3). The LLM is provided with the input of a description of a molecule and a set of QAs is generated.

- In Figures 4 and 5, we depict the prompt used for validating the generated data with an LLM(section 3.4). In the first stage, the LLM is provided with the generated question, answer, and molecule description, and tasked with inferring the correct option based on this input. In the next stage, the LLM is given the filtered questions and prompted to evaluate their relevance in relation to the molecular characteristics.

- In Figures 6 and 7, we discuss the prompts used for inference from LLMs(section 4.3), and also for fine-tuning LLMs(section 4.4).

---

**System Text:**
""" Your task is to generate a set of questions about a molecule given its description. Each question should contain 5 multiple choice options. The correct answer should be an integer between 1 and 5. Also categorize each question into 1. Physical Properties, 2. Chemical structure information/properties 3. Biological or therepeutic information 4. Origin/Molecule Synthesis, 5. Applications.
Follow these strict rules:
1.All questions should be factually grounded in the caption, using the same terminology, and do not include any information not present in the caption. The focus of the questions should be the molecule.
2.The options should be thematically similar but should be discriminative enough and EXACTLY one of the options is correct. Avoid options like "all" or "none".
3.Anonymize the actual name of the molecule in the QAs, and refer to it as " molecule".
The output should only include json parsable text (with no additional text), in the following format:
[{"question 1":question, "options":[option1, option2,..], "correct":1, "sentence": The molecule is option1, "category":2 }, {"question": .., .. :.. },..]"""
**User Text:**
""" Acetone is a manufactured chemical that is also found naturally in the environment. It is a colourless liquid with a distinct smell and taste. It evaporates easily, is flammable, and dissolves in water. It is also called dimethyl ketone, 2-propanone, and beta-Keto propane. """

---

Figure 3: **Prompt used for QA generation:** The LLM is provided with the input of a description of a molecule and is prompted to generate a set of QAs is generated

System Text:
""" You are given a description about molecule and a list of questions following with multiple choice options. For each question, provide the index of a single correct option. If the answer cannot be inferred from the description or the correct option is not available, output 0 for that question. The output be a list of integers (between 1 to 5) and strictly do not include any other text before or after the answer.
Example Output: [answer_idx_for_q1, answer_idx_for_q2, ....].
"""

User Text:
""" Description: Acetone is a manufactured chemical that is also found naturally in the environment. It is a colourless liquid with a distinct smell and taste. It evaporates easily, is flammable, and dissolves in water. It is also called dimethyl ketone, 2-propanone, and beta-Keto propane.
Questions:
Question 1: What is an alternative names for the molecule?
Options: ["Isopropanol", "Methyl ethyl ketone", "Beta-Keto propane", "Toluene", "Acetic acid" ]
Question 2:  What is the evaporation characteristic of the molecule?
Options: [ "Does not evaporate", "Evaporates under high temperature", "Evaporates easily", "Sublimates instead of evaporating", "Evaporates only in vacuum" ] """

Figure 4: **Prompt used for validation:** The LLM is provided with a a generated Question and options, and the description of the question it was generated from. The LLM is then prompted to identify the correct option.

System Text:
""" You are given a description about a molecule, and a set of question-answer pairs generated from it using an LLM. The objective of the questions is to test the understanding of the molecule and its properties (such as structure, manufacturing, chemical/physical properties, biological applications etc). Your task is to filter out any questions that are not relevant to the molecule, or that are not useful for testing the understanding of the molecule. Also provide a brief explanation for each question you filter out.
"""

User Text:
""" Description: Acetone is a manufactured chemical that is also found naturally in the environment. It is a colourless liquid with a distinct smell and taste. It evaporates easily, is flammable, and dissolves in water. It is also called dimethyl ketone, 2-propanone, and beta-Keto propane.
Questions:
Question 1: What is an alternative names for the molecule? Answer: "Beta-Keto propane"
"""

Figure 5: **Prompt used for validation step 2:** The LLM is then provided with the questions from the previous stage and tasked with evaluating their relevance in the context of deciphering molecular characteristics

**Prompt for Inference (or) Finetuning:**
""" You are given a SMILES string of a molecule, a question about the molecule and a set of candidate options.
Output the index of the option that best answers the question. Do not include additional text in the output.
Question: What is an alternative names for the molecule?
SMILES string: CC(=O)C
Options: (1) "Isopropanol" (2) "Methyl ethyl ketone" (3) "Beta-Keto propane" (4) "Toluene" (5) "Acetic acid"
"""

Figure 6: **Prompt for Molecule QA inference:** The LLM is provided with an input SMILES string, and a set of options, and is prompted to identify the correct option.

**Prompt for Inference (or) Finetuning:**
""" You are given a sentence describing a molecule. Choose the SMILES string that best describes the sentence.
Output the index of the best correct option only and nothing else. Do not include additional text in the output.
Sentence: The molecule is also called as Beta-Keto propane
Options: (1) C(=O)O (2) CCC(=O)C (3) CC(C)O (4) CC(=O)C (5) CC(=O)O """

Figure 7: **Prompt for Molecule Retrieval inference:** The LLM is provided with an input sentence about a SMILES string, and a set of options, and is prompted to identify the correct SMILES seqeuce.

## Appendix E. Dataset Categories and Examples

1. **Chemical Information** covers the chemical structure, functional groups, and chemical properties.

2. **Physical Properties** addresses the properties such as solubility, physical state, and odor.

3. **Biological Information** contains the molecules' role in biological pathways, drug applications, and drug toxicity.

4. **Source** details the molecules' origin and manufacturing processes.

5. **Application** describes application areas such as perfumes, fertilizers, and insecticides.

**Examples:**

- What is the appearance of the molecule?

  – **Options:** lustrous, brittle, silvery solid; dull, flexible, black solid; shiny, flexible, gold solid; dull, brittle, gray solid; shiny, brittle, silver solid
  – **Category:** Physical properties

- What is the molecule's pH level?

  – **Options:** acidic; basic; neutral; alkaline
  – **Category:** Chemical Information

- What type of immunity does the molecule affect?

  – **Options:** Humoral Immunity; Cell-mediated Immunity; Innate Immunity; Adaptive Immunity; Passive Immunity
  – **Category:** Biological Information

- How is the molecule manufactured?

  – **Options:** By reacting benzene with hydrochloric acid; By reacting ethylene with sulfuric acid; By reacting acetylene with nitric acid; By reacting methane with sulfuric acid; By reacting toluene with concentrated sulfuric acid
  – **Category:** Manufacturing/Source

- What is the physical state of the molecule at room temperature?

  – **Options:** solid; liquid; gas
  – **Category:** Physical Properties

- What is the source of the molecule?

  – **Options:** Arabidopsis thaliana; Ludwigia repens; Escherichia coli; Streptomyces coelicolor
  – **Category:** Manufacturing/Source

- What is the molecule's common use in the food industry?

  - **Options:** preservative; flavor enhancer; texture modifier; coloring agent
  - **Category:** Applications

- How has the EPA classified the molecule?

  - **Options:** Group A, human carcinogen; Group B, probable human carcinogen; Group C, possible human carcinogen; Group D, not classifiable as to human carcinogenicity; Group E, evidence of non-carcinogenicity
  - **Category:** Biological Information

- What is the flash point of the molecule?

  - **Options:** 100 °F; 150 °F; 175 °F; 200 °F; 250 °F
  - **Category:** Physical Properties

- What is the molecule's relationship to phenols?

  - **Options:** It is a conjugate acid of a phenolate; It is a human xenobiotic metabolite; It is a mouse metabolite; It has a role as a disinfectant; It is an antiseptic drug
  - **Category:** Chemical Information

## Appendix F. Dataset efficacy evaluation

To assess the overall reliability of our dataset, we conducted a benchmark by randomly sampling 400 data points from the test split. The data points were evaluated on three criteria: whether the question could be logically derived from the provided caption, the unambiguous correctness of the answer, and the relevance of the question—specifically, ensuring it avoids uninformative queries and contributes to meaningful chemical or biological insights based on the structure. Out of the sampled data, 391 points met these criteria. To understand the implications of this result for the entire dataset, we calculated the p-value using a hypergeometric distribution[1]. A hypergeometric test is used to measure the probability of obtaining a specific number of successes in a given number of draws from a finite population containing a certain amount of successes. With parameters k=400, n=391, N=11,922 (i.e the size of all test split), and K=0.961*11,922, we found a p-value of <0.05. This result suggests that the dataset is over 96.1 percent accurate, demonstrating a high level of reliability. Details of the random test samples used for this evaluation can be accessed through the project's repository.

To illustrate our evaluation process, we present representative examples of both accepted and rejected cases:

### F.1 Accepted Examples

- **PubChem ID: 21580808**
  **Question:** What is the molecule resulting from?
  **Options:**

  1. Protonation of the oxygen of the primary amino group of sotalol

  2. Protonation of the nitrogen of the secondary amino group of sotalol

  3. Deprotonation of the nitrogen of the primary amino group of sotalol

  4. Protonation of the oxygen of the secondary hydroxyl group of sotalol

  5. Deprotonation of the oxygen of the primary hydroxyl group of sotalol

  **Accepted because:** The question addresses specific chemical modifications with clearly distinguishable options.

- **PubChem ID: 47528**
  **Question:** What is the mechanism of action of the molecule on vascular smooth muscles?
  **Options:**

  1. Membrane depolarization

  2. Membrane hyperpolarization

  3. Increased transmembrane sodium conductance

  4. Increased intracellular concentration of cyclic AMP

  5. Reduced transmembrane potassium conductance

---

1. See the SciPy documentation for the hypergeometric distribution: `https://docs.scipy.org/doc/scipy/reference/generated/scipy.stats.hypergeom.html`

**Accepted because:** The question relates to structure-function relationships with distinct, non-overlapping answer choices.

- **PubChem ID: 1711945**
  **Question:** Where is the molecule naturally found?
  **Options:**

  1. Tilia platyphyllos

  2. Tilia tomentosa

  3. Sargassum natans

  4. Sargassum micracanthum

  5. Sargassum flavescens

  **Accepted because:** The question has a single, verifiable correct answer among distinct options.

## F.2 Rejected Examples

- **PubChem ID: 54671008**
  **Question:** When did the molecule receive FDA approval?
  **Options:**

  1. 10 October 2006

  2. 12 October 2007

  3. 12 October 2008

  4. 10 October 2009

  5. 12 October 2010

  **Rejection Rationale:** Relies on temporal metadata information rather than molecular properties, which cannot be inferred from structure.

- **PubChem ID: 10129**
  **Question:** What type of odor does the molecule have?
  **Options:**

  1. Strong

  2. Mild

  3. Sweet

  4. Pungent

  5. Unpleasant

  **Rejection Rationale:** Options lack clear differentiation and are potentially overlapping.

- **PubChem ID: 101562486**
  **Question:** What is the general class of biomolecules to which the molecule belongs?
  **Options:**

1. Carbohydrate

2. Lipid

3. Oligopeptide

4. Nucleic acid

5. Heterocycle

**Rejection Rationale:** Multiple options could be technically correct.

- **PubChem ID: 53361968**
  **Question:** What characteristic may make the molecule a desirable therapy?
  **Options:**

  1. It is less expensive

  2. It is less likely to generate resistance

  3. It is only for treatment-naive patients

  4. It is only for PI-experienced patients

  5. It is only for HIV-2 infections

**Rejection Rationale:** Addresses clinical outcomes not directly inferrable from structure.

## Appendix G. Analysis of Generalization of Finetuned models

We construct an *out-of-distribution (OOD) test split* to assess model transferability under distribution shift.

### G.1  Construction of the OOD Test Split

To simulate an OOD setting while maintaining sufficient coverage and balance, we adopt the following procedure:

1. **Sentence-level embedding:** For each question–answer pair in the test set, we concatenate the question and its gold answer into a single string. We then encode this using the `all-MiniLM-L12-v2` Sentence-Transformer (384 dimensions).

2. **Distance to training data:** For every encoded test point, we compute its cosine distance to the closest question–answer pair in the training set.

3. **Selecting the OOD split:** We sort the test points by distance and retain the top 10% most distant examples as our OOD slice. These represent samples least similar to those observed during training, and thus serve as a proxy for generalization assessment.

### G.2  Results

| Model | Full Test | OOD Slice |
|---|---|---|
| MoleculeSTM | 65.14 | 64.38 |
| MoMu | 65.08 | 64.92 |
| LLaMA-3 8B | 60.41 | 61.07 |
| LLaMA-2 7B | 41.84 | 42.58 |
| Galactica-125M | 43.97 | 45.50 |
| Galactica-1.3B | 60.98 | 62.36 |
| Galactica-6.7B | 69.01 | 69.41 |
| MolT5-large | 34.15 | 38.01 |
| MolT5-large-s2c | 34.69 | 34.16 |
| Qwen-0.6B (GRPO) | 53.75 | 53.95 |
| Qwen-4B (GRPO) | 63.97 | 62.89 |

Table 7: Accuracy of models on the full test set vs. the OOD slice for Molecule QA.

Across both QA and Retrieval tasks, we observe that most models perform comparably on the OOD slice relative to the full test set, indicating a reasonable degree of generalization. Interestingly, some models (e.g., Galactica-1.3B, LLaMA-3 8B) show marginally better performance on the OOD slice, which could be attributed to their robustness to semantic variation in the input.

| Model | Full Test | OOD Slice |
|---|---|---|
| MoleculeSTM | 65.27 | 20.51 |
| MoMu | 63.60 | 20.42 |
| LLaMA-3 8B | 20.60 | 22.85 |
| LLaMA-2 7B | 20.58 | 23.86 |
| Galactica-125M | 21.62 | 20.33 |
| Galactica-1.3B | 22.17 | 20.42 |
| Galactica-6.7B | 22.30 | 22.16 |
| MolT5-large | 23.54 | 21.43 |
| MolT5-large-c2s | 23.00 | 22.80 |
| Qwen-0.6B (GRPO) | 29.49 | 12.50 |
| Qwen-4B (GRPO) | 58.87 | 60.00 |

Table 8: Accuracy of models on the full test set vs. the OOD slice for Molecule Retrieval.

## Appendix H. Baseline Models, Training, and Finetuning

In this section, we expand on the specific details of different models used to evaluate the MolQA dataset.

### H.1 Model Details

#### H.1.1 SINGLE ENCODER ARCHITECTURES:

**Scibert**: SciBERT (Beltagy et al., 2019) is an encoder model, that leverages a pre-trained BERT framework, subsequently fine-tuned on a substantial corpus of scientific papers, predominantly from the biomedical domain (constituting 85% of its training data). This specialization makes SciBERT a useful baseline for our study. It has a robust performance across various scientific tasks, including named entity recognition and text classification.

**KV-PLM**: KV-PLM(Zeng et al., 2022) is a single encoder model derived from SciBERT, further trained on molecule-text pairs sourced from PubChem. The training process begins with pre-training, during which SMILES sequences are appended to molecular descriptions to form the training data. The model employs a masking strategy where certain tokens representing both molecular structures and biomedical text are masked at random. The model's task is to predict these masked tokens based on the surrounding context. Following pre-training, KV-PLM undergoes fine-tuning for text retrieval tasks. In this phase, the model learns to accurately retrieve specific text descriptions based on SMILES sequence inputs, utilizing a max hinge loss function. This loss is given by:

$$
\begin{aligned}
\mathcal{L}_{\text{MH}} = & \max_{\mathbf{d}'} \left[ \alpha + \text{s}\left(\mathbf{m}, \mathbf{d}'\right) - \text{s}(\mathbf{m}, \mathbf{d}) \right] \\
& + \max_{\mathbf{m}'} \left[ \alpha + \text{s}\left(\mathbf{m}', \mathbf{d}\right) - \text{s}(\mathbf{m}, \mathbf{d}) \right],
\end{aligned}
\tag{1}
$$

here $\mathcal{L}_{\text{MH}}$ represents the max hinge loss, $\mathbf{m}$ and $\mathbf{d}$ denotes the molecule (SMILES sequence) and its corresponding text description, respectively. The terms $\mathbf{d}'$ and $\mathbf{m}'$ refer to a negative text and molecule that do not match the original pairing and the function $\text{s}(\mathbf{m}, \mathbf{d})$ calculates the similarity score between a molecule and a document.

#### H.1.2 MULTIMODAL ARCHITECTURES:

**MoleculeSTM:** The paper (Liu et al., 2023a) introduces a framework that uses a dual-encoder to extract and align representations of text and molecules. The framework employs a Graph Isomorphism Network (GIN) (Xu et al., 2018) to encode chemical data, represented by $f_c$, and SciBert to encode textual data, denoted as $f_t$. The GIN model is initialized from GraphMVP (Liu et al., 2021), which does multi-view pretraining between the 2D topologies and 3D geometries from the GEOM dataset (Axelrod and Gomez-Bombarelli, 2022). The components are trained end-to-end on a dataset that contains molecules and their descriptions sourced from PubChem. The model's learning process is governed by the InfoNCE loss:

$$
\begin{aligned}
\mathscr{L}_{\text{InfoNCE}} = -\frac{1}{2}\mathbb{E}_{\boldsymbol{x}_c, \boldsymbol{x}_t} & \left[ \log \frac{\exp\left(E\left(\boldsymbol{x}_c, \boldsymbol{x}_t\right)\right)}{\exp\left(E\left(\boldsymbol{x}_c, \boldsymbol{x}_t\right)\right) + \sum_{\boldsymbol{x}_{t'}} \exp\left(E\left(\boldsymbol{x}_c, \boldsymbol{x}_{t'}\right)\right)} \right. \\
& \left. + \log \frac{\exp\left(E\left(\boldsymbol{x}_c, \boldsymbol{x}_t\right)\right)}{\exp\left(E\left(\boldsymbol{x}_c, \boldsymbol{x}_t\right)\right) + \sum_{\boldsymbol{x}_{c'}} \exp\left(E\left(\boldsymbol{x}_{c'}, \boldsymbol{x}_t\right)\right)} \right]
\end{aligned}
\tag{2}
$$

Here $\boldsymbol{x}_c$ and $\boldsymbol{x}_t$ represent the input chemical structure and textual description, respectively. $f_c$, $f_t$ represent the chemical and text representation model, and $p_c$, $p_t$ represents chemical, text projection matrices. The function $E(\boldsymbol{x}_c, \boldsymbol{x}_t) = \langle p_c \circ f_c(\boldsymbol{x}_c), p_t \circ f_t(\boldsymbol{x}_t) \rangle$ calculates the similarity. The goal is to distinguish between correctly matched chemical-text pairs $(\boldsymbol{x}_c, \boldsymbol{x} * t)$ and mismatched pairs $(\boldsymbol{x}c, \boldsymbol{x}t', \boldsymbol{x} * c', \boldsymbol{x}_t)$, enhancing the model's ability to map chemical structures to their descriptive texts accurately.

**MoMu:** MoMu (Su et al., 2022) is another dual-encoder model similar to MoleculeSTM, leveraging both SciBERT for textual data and a Graph Isomorphism Network (GIN) for chemical structures. Similar to MoleculeSTM, MoMu employs a GIN network but is initialized with random weights, which are then trained using a contrastive loss mechanism akin to that used in MoleculeSTM. This baseline comparison underscores the potential enhancements 3D pretraining brings to the model's ability to capture complex molecular structures.

### H.1.3 LARGE LANGUAGE MODELS

**Llama**: Llama (Touvron et al., 2023) is a series of decoder-only, autoregressive transformer models trained on a large general corpus. Llama demonstrates exceptional performance in various tasks, including common sense reasoning, closed-book QA, mathematical reasoning, and code generation. Llama has been fine-tuned on a select instructional dataset to follow human instructions effectively. Given its extensive application range, assessing Llama's chemical understanding capabilities is of interest. We have experimented with the smallest and the largest versions of the Llama-2 and Llama-3 series of models.

**GPT 3.5 Turbo** We further benchmark the GPT-3.5 Turbo (OpenAI, 2022) model, a member of the Generative Pre-trained Transformer series.

**Galactica**: We also benchmark Galactica (Taylor et al., 2022), a set of scientific language models that are autoregressive, decoder-only similar to the previous models. These models are trained to recognize and understand a wide range of scientific information, such as chemical structures represented by SMILES strings, sequences of amino acids, computer code, and mathematical equations. The dataset used contains a large collection of scientific documents and research papers. It is shown to be effective on specialized biomedical datasets like PubMedQA and MedMCQA. For this study, we use Galactica models of different sizes, with 125M, 1.3B, and 6.7B parameters.

### H.1.4 ENCODER-DECODER MODELS

**MolT5:** MolT5 (Edwards et al., 2022) is an encoder-decoder model, built by fine-tuning a T5 (Raffel et al., 2020) model. The model is trained in two stages. First, the the model is trained with masked language modeling objective, to encode and decode SMILES string and molecule captions. Next, the model is fine-tuned to generate SMILES strings or captions from the captions or the SMILES strings input respectively. The dataset used for fine-tuning is CheBI-20 (Edwards et al., 2021)).

**BioT5:** BioT5 (Pei et al., 2023) is an encoder-decoder model based on the T5 (Raffel et al., 2020) architecture, expanding on MolT5 by replacing SMILES with SELFIES representations, making molecular generation more robust and reliable. Additionally, BioT5 integrates proteins into its framework, enabling advanced multi-modal capabilities. The model is trained on a diverse set of paired modalities, including protein FASTA sequences, wrapped molecule-text pairs, and molecule-text captions. This comprehensive multi-modal approach allows BioT5 to handle a wide range of tasks, bridging textual, molecular, and protein-based representations effectively.

**BioT5+:** BioT5+ Pei et al. (2024) improves upon BioT5 by incorporating additional data sources and task-specific fine-tuning while maintaining a similar training methodology. The model is trained on a broader range of tasks, including molecular property prediction, retrosynthesis, and protein-protein interaction (PPI) prediction. These additional capabilities, combined with its robust multi-modal training approach, make BioT5+ a powerful tool for tackling complex challenges in molecular and protein sciences.

### H.2 Zero-shot Inference:

**Large Language Models:** For the Llama series of models, zero-shot inference was performed using the API service offered by Microsoft Azure AI services. For GPT-3.5, zero-shot inference is performed using the OpenAI platform. For Galactica, model weights are obtained from the Hugging Face library.

**Multi-modal Architectures:** For SciBERT and KV-PLM, the model weights are initialized from the MoleculeSTM official repository: `https://github.com/chao1224/MoleculeSTM`. We retrained MoMu and MoleculeSTM, due to potential data leakage issues by removing the samples in the pertaining set that overlap with the test set. 30 Epochs of training on the pre-training dataset were performed for MoleculeSTM and MoMu, which took about 40 hours on an NVIDIA RTA A6000 GPU. The code and model parameters were obtained from Molecule STM's official repository. THe learning rate used was 1e-5 and a batch size of 45. For MoMu, we used the augmentation probability of 0.2. A temperature of 0.1 is used for both the models.

### H.3 Finetuning:

**Galactica:** 3 Epochs of training on the finetuning dataset were performed by the other models. LoRA (Hu et al., 2021) was used to finetune the query and value vectors of Galactica 125M, 1.3B, and 6.7B. Training and evaluation took around 2.5 hours, 5 hours, and 30 hours respectively on a NVIDIA A100 80GB PCIe. The learned rate used was 2e-5, a weight decay of 0.01, and a batch size of 8 using Huggingface's Trainer for causal language modeling (because the base OPT model is a decoder-only model)[2]. The LoRA parameters are $r = 16$, $\alpha = 32$ and a lora_dropout of 0.05.

**MoleculeSTM and MoMu:** The finetuning was performed in the same setting as training, as described in Section H.2. Both the models are fine-tuned for 3 epochs, at a learning rate of 1e-15. The total finetuning time is about 1 hour.

**Llama2-7B and Llama3-8B models:** Both the models are trained for 3 epochs using LoRA on an NVIDIA A100 80GB PCIe. The total training time is approximately 30 hours for each model. The Lora parameters used are $r = 16$, $\alpha = 32$ and a lora_dropout of 0.1. The learning rate is 1e-5 and the weight decay used is 1e-4.

**MolT5, BioT5 and BioT5 plus:** The models are finetuned for 10 epochs on an NVIDIA A100 80GB PCIE. The approximate training time for the MolT5 model is 3.5 hours, while it is 6 hours for the BioT5 models. These models are trained with a learning rate of 2e-5, a weight decay of 0.01 and a batch size of 8 using the Huggingface's trainer.

**Qwen-0.6B and Qwen-4B (GRPO):** The models are fine-tuned for 2 epochs using GRPO with LoRA applied via the Unsloth library. LoRA parameters are set to $r = 8$ and $\alpha = 16$. For both Molecule QA and Molecule Retrieval tasks, the model is prompted to end its output with the format

---

2. `https://huggingface.co/docs/transformers/en/tasks/language_modeling`

"The final answer is... option" following a reasoning chain. A reward of +1 is assigned if the correct option appears in the final answer, and 0 otherwise. Training was performed on 2 NVIDIA A100 80GB GPUs. The approximate training time is 12 hours for the 0.6B model and 60 hours for the 4B model.

### H.4 A note on 3DMoLLM

Additionally, we attempted to fine-tune the BLIP-like model, as proposed in the 3DMoLLM paper (Li et al., 2024), on the MolTextQA dataset. This model methodology involves projecting the 3D structure of a molecule into the space of LLM tokens and utilizing these tokens for text generation. Unfortunately, we encountered challenges during fine-tuning using the default configurations provided in the code repository. Specifically, we were unable to finetune the model to generate answers in the required format, which impeded our ability to perform any meaningful analysis.

