# OpenReview forum: "MolTextQA: A Question-Answering Dataset and Benchmark for Evaluating Multimodal Architectures and LLMs on Molecular Structure–Text Understanding"
_DMLR — Accepted by DMLR_

### Review · Reviewer_2KWL · 2025-04-19

**Recommendation:** 3
**Confidence:** 1

**Summary Of Contributions:**

This paper introduces MolTextQA, a large, curated dataset containing nearly 500,000 question-answer (QA) pairs covering 240,000 molecules from PubChem, aimed at advancing molecular structure-to-text relationship learning. The dataset supports multiple-choice question answering and molecule retrieval task, and covers five major categories which are chemical information, physical properties, biological information, source, and application. The QA pairs were generated using large language models (LLMs), including LLaMA3-70B, LLaMA3-8B, and validated by GPT-3.5, followed by evaluation through human-annotated random sampling, with results assessed via statistical analysis. Moreover, the authors benchmarked a variety of model architectures under zero-shot and fine-tuned settings. The dataset, models, and accompanying code are publicly available.

**Strengths:**

The dataset is thoroughly validated using a two-stage LLM pipeline and human checks, achieving >96% factual accuracy. The authors benchmark a broad set of models, from multi-modal architectures to LLMs, providing a comprehensive and insightful comparison across QA and retrieval tasks. The paper is clearly presented, well-motivated, and relevant to research community.

**Audience:**

Yes

**Broader Impact Concerns:**

The authors have included a broader impact statement that addresses potential applications in drug discovery, retrosynthesis, and materials science. While their existing discussion is adequate, it could be further enhanced by elaborating on ethical considerations related to the potential misuse or biases inherent in dataset use is strongly recommended.

**Claims And Evidence:**

The claims made in the submission are supported by evidence. The methodologies for dataset creation, validation, and benchmarking are sound. The statistical validation of dataset quality (e.g., hypergeometric test with p<0.05) provides convincing evidence of reliability.

**Datasets And Benchmarks:**

The dataset is well-documented, with clear structure, licensing, usage details, and public availability with DOI. Benchmarking is well-described, supporting reproducibility.

**Extended Submissions:**

The submission does not appear to be an extended version of a prior work.

**Limitations:**

The authors have appropriately addressed the limitations of their work, including minor dataset inaccuracies and the presence of some simple questions. Additional considerations would strengthen the paper:

1. The dataset uses molecules from PubChem which may introduce potential biases in terms of which molecules are well-documented. This could potentially lead to models that perform better on common or well-studied molecules but struggle with novel or less-documented compounds.

2. The authors mention potential applications in drug discovery and materials science, they could provide a more explicit discussion of the ethical implications and broader societal impact of their work. And it would benefit from exploring potential negative societal impacts related to dataset misuse. Documenting these risks and proposing mitigation strategies would demonstrate more comprehensive ethical consideration and responsible research practices.

**Requested Changes:**

The dataset relies on LLMs (LLaMA3, GPT-3.5) for generating and validating question-answer pairs. However, the risk of hallucinated facts or inherited biases from these models is not discussed. Discuss common failure modes observed during QA generation (e.g., hallucinations, speculative answers) and how the authors mitigate them would strengthen this point.

**Strengths And Weaknesses:**

Strengths:
1. The dataset offers a valuable contribution to molecular structure–text relationship learning. It is designed to prevent information leakage by anonymizing molecule names, thereby encouraging models to rely on structural features rather than memorized associations.

2. Extensive benchmarking across a range of model architectures is presented, both in zero-shot and fine-tuned settings. This provides useful comparative insights into the relative strengths and limitations of different models for structure-text QA tasks.

3. The authors publicly release all dataset construction details, including prompts and benchmarking code. The dataset is distributed under a permissive CC BY 4.0 license and is assigned a DOI, supporting reproducibility across the research community.

Weaknesses:
1. The dataset focuses solely on multiple-choice questions derived from sentence-level descriptions. This may restrict the depth of reasoning required and limit the dataset’s ability to fully assess cross-modal understanding between molecular structures and textual data.

2. Since the dataset was initially generated using large language models such as LLaMA, there remains a risk of hallucinated or subtly inaccurate information. Although a two-stage validation process and human annotation were employed, concerns about factual reliability remain—especially for less common molecules.

3. Fine-tuning experiments were conducted on a relatively small subset of the data. As a result, the reported performance may underestimate the capabilities of large models. This is particularly relevant given the rapid advancements in large language models, where newer generations exhibit significantly improved learning efficiency and cross-modal reasoning capabilities. A more thorough fine-tuning regimen could potentially offer stronger performance and more robust conclusions.

---

### Review · Reviewer_RdsX · 2025-05-15

**Recommendation:** 3
**Confidence:** 1

**Summary Of Contributions:**

The paper introduces a novel dataset containing 500,000 question-answer pairs covering 240,000 molecules from PubChem. This dataset can be used for structure directed questions and text-based molecule retrieval. The authors provide extensive experimental evaluation, addressing how LLMs and multimodal architectures perform in Molecule QA and Molecule Retrieval tasks.

**Strengths:**

I think this work is relevant as the proposed dataset could benefit the research community. The authors also discuss some of the limitations of the current work, hence this helps the usability of the provided assets. I argue this can become an impactful contribution.

**Audience:**

Yes

**Broader Impact Concerns:**

The paper provides a Broader Impact section, which I think addresses most of the implications.

**Claims And Evidence:**

The paper supports its claim with an extensive empirical evaluation.

**Datasets And Benchmarks:**

The paper provides a link to huggingface where the dataset is stored,

**Extended Submissions:**

N/A

**Limitations:**

The paper discusses limitations of the current work.

**Requested Changes:**

The main requested change is the inclusion of some uncertainty measures in the emprical evaluation.

While I understand retraining is not possible for these large models, one could exploit bootstrap approaches to provide some measures of uncertainty regarding the results (see e.g., [1]).
This would improve the overall empirical evaluation.

[1] - Rajkomar, A. et al. (2018). Scalable and accurate deep learning with electronic health records. NPJ digital medicine, 1(1), 18.

**Strengths And Weaknesses:**

The paper's main strengths are:

- the paper is clearly written, with sufficient details;
- the work's motivation is relevant;
- the empirical evaluation provide interesting insights on the ability of LLMs;


The paper's main weaknesses are:

- the empirical evaluation does not contain any measure of uncertainty (see Requested changes comments);

---

### Review · Reviewer_Rm9W · 2025-06-05

**Recommendation:** 3
**Confidence:** 1

**Summary Of Contributions:**

This paper introduces MolTextQA, a large-scale benchmark dataset and evaluation suite tailored for molecular structure-text reasoning tasks. The dataset comprises over 500,000 multiple-choice QA pairs derived from PubChem for 240,000+ unique small molecules, covering a wide spectrum of properties and applications. The questions are categorized into five types (chemical, physical, biological, source, and application) and are validated through a two-stage process involving both LLM-based and human verification, yielding a reported accuracy exceeding 96.1%. Extensive benchmarking is conducted on state-of-the-art multimodal and large language models (LLMs), and both the dataset and fine-tuned models are released publicly.

**Strengths:**

1. The dataset and model are publicly released.
2. This dataset contains 500,000 question-answer pairs covering 240,000 molecules from various domains, such as chemical and physical, which is on a large scale. This contributes to the AI4Sci field.
3. The zero-shot results of contemporary LLMs, encoder-decoder models and separate encoder models are poor, and the fine-tuned models's performance gets improved, demonstrating the need of more powerful models in enhancing the knowledge in this field.

**Audience:**

Yes

**Claims And Evidence:**

This submission makes claims clearly.

**Datasets And Benchmarks:**

This dataset provides adequate information regarding data collection and organization and is publicly available.

**Extended Submissions:**

This work is not an extended version of a previously published work.

**Limitations:**

1. The models used in the evaluation are somewhat outdated. It is suggested to include more state-of-the-art open-source and proprietary LLMs and LRMs, such as the Qwen2.5 and Qwen3 series, QwQ-32B, GPT-4o, GPT-4.1, OpenAI-o3, and OpenAI-o4-mini.
2. The dataset is categorized into five distinct types: Chemical Information, Physical Properties, Biological Information, Source, and Application. However, the evaluation only involves fine-tuning the models on the full training set that includes questions from all five categories. The study lacks an analysis of model transferability, specifically, how well a model fine-tuned on one category performs when tested on another.
3. The dataset currently consists only of multiple-choice questions. Including open-ended questions would enrich the dataset and provide a more comprehensive evaluation of model capabilities.

**Requested Changes:**

1. Involve more recent models.
2. Conduct a deep analysis, such as model transferability analysis. Currently, only some simple zero-shot and fine-tuning experiments are included.
3. It would be better if some open-ended questions are included. If not, it is fine.

Overall, this dataset is solid and comprehensive and significantly advances the field of molecular structure–text reasoning.
However, the current evaluation protocol could be further enriched. Specifically, while the benchmark includes a variety of fine-tuned models, it would benefit from the inclusion of more recent and diverse state-of-the-art LLMs and LRMs (large reasoning models), especially models that have demonstrated strong performance in related multimodal or scientific reasoning tasks. In addition, it would be better if deeper analysis such as model generalization.

**Strengths And Weaknesses:**

**strengths**
1. The dataset and model are publicly released.
2. The dataset is large-scale and comprehensive.
3. Extensive experiments are conducted.

**weaknesses**
1. The involved models are a little bit too old. The evaluation on more state-of-the-art open-source and close-source LLMs and LRMs are needed, such as Qwen2.5 series, Qwen3 series QwQ-32B, GPT-4o, GPT-4.1, OpenAI-o3, OpenAI-o4-mini.
2. Only simple zero-shot and fine-tuning results are reported. Deeper analysis is required.
3. The dataset currently consists only of multiple-choice questions. Including open-ended questions would enrich the dataset and provide a more comprehensive evaluation of model capabilities.